# The Multifaceted Role of FUT8 in Tumorigenesis: From Pathways to Potential Clinical Applications

**DOI:** 10.3390/ijms25021068

**Published:** 2024-01-15

**Authors:** Meng Shi, Xin-Rui Nan, Bao-Qin Liu

**Affiliations:** Department of Biochemistry & Molecular Biology, School of Life Sciences, China Medical University, Shenyang 110122, China; sm1233690317@163.com (M.S.); 2022122357@cmu.edu.cn (X.-R.N.)

**Keywords:** FUT8, core fucosylation, malignancy, pathway, tumor

## Abstract

FUT8, the sole glycosyltransferase responsible for N-glycan core fucosylation, plays a crucial role in tumorigenesis and development. Aberrant FUT8 expression disrupts the function of critical cellular components and triggers the abnormality of tumor signaling pathways, leading to malignant transformations such as proliferation, invasion, metastasis, and immunosuppression. The association between FUT8 and unfavorable outcomes in various tumors underscores its potential as a valuable diagnostic marker. Given the remarkable variation in biological functions and regulatory mechanisms of FUT8 across different tumor types, gaining a comprehensive understanding of its complexity is imperative. Here, we review how FUT8 plays roles in tumorigenesis and development, and how this outcome could be utilized to develop potential clinical therapies for tumors.

## 1. Introduction

Fucosylation is one of the most widely and frequently occurring glycosylation modification in vivo. Analysis of the carbohydrate structure of 3299 mammals in the GLYCOSCIENCES.de databank revealed that L-fucose accounts for 7.2% and 23.8% in the abundance of monosaccharides and terminal monosaccharide moieties, respectively [1], indicating that fucose is a crucial and widespread component in glycan modification. Fucosyltransferases (FUTs) facilitate the transfer of L-fucose, obtained from Guanosine 5′-diphospho-β-L-fucose (GDP-fucose), to oligosaccharide chains. So far, 13 fucosyltransferases in mammals have been identified, including N-fucosyltransferases (FUT1-11) and peptide-O-fucosyltransferases (POFUT1,2), as shown in Table 1. Notably, fucosyltransferase 8 (FUT8) is the only enzyme responsible for the addition of fucose to the first N-acetylglucosamine (GlcNAc) residue adjacent to peptide chains in N-glycans, catalyzing the formation of an α-(1,6)-linkage [2,3].

FUT8, a distinctive member of the fucosyltransferase family, was the latest one to be discovered [2,17]. However, its role and impact on the body cannot be overlooked. FUT8 has been demonstrated to be involved in numerous physiological and pathological processes, including blood coagulation [18] and cystic fibrosis [19]. And its implication in tumors has gradually become the focus of research. In particular, core fucosylated α-fetoprotein (AFP), detected by Lens culinaris agglutinin (LCA) affinity, exhibits presence in tissues prior to tumor formation, making it a valuable diagnostic marker [20,21]. The exponential growth of publications indexed in the Web of Science, as depicted in Figure 1, illustrates the remarkable progress in FUT8 research over the past two decades.

## 2. Biological Characteristics of FUT8

### 2.1. Gene Structure and Expression Distribution of Human FUT8

The human FUT8 gene is located on chromosome 14q23.3, distinctly separate from other fucosyltransferases, and consists of 387,280 bases. Due to alternative splicing or alternative promoter usage, it gives rise to 17 transcripts, and the final expression product contains 575 amino acids [22]. The analysis of the FUT8 promoter region revealed the presence of a typical TATA-box sequence, but not a CCAAT motif [23]. FUT8 is abundantly expressed in multiple organs like the human brain, bladder, stomach, small intestine, and salivary glands. Studies on gene regulation for post-translational modification have demonstrated that the fucosylation of transferrin and immunoglobulin G is regulated by distinct protein-specific variants of FUT8 and FUT6 [24]. As a result, different FUT8 variants may catalyze core fucosylation of distinct proteins, showing tissue-specific regulatory patterns by influencing the binding of specific transcription factors. This finding provides insight into the existence of multiple FUT8 variants.

### 2.2. Protein Structure of FUT8

Due to its indispensable role in bodily functions, the crystal structure of the FUT8 protein was analyzed to gain insight into its molecular mechanism of action. It is shown that FUT8 is a multi-domain enzyme, which comprises of 15 strands and 16 helices, consisting of an N-terminal coiled-coil domain, a GT-B folded domain, and a C-terminal Src homologous domain 3 (SH3) domain [25]. The GT-B domain is the catalytic domain containing a DXD motif that does not require metal ions to maintain enzyme activity. However, FUT8 contains only one Rothman fold unlike other GT-B type glycosyltransferases. The N-terminal coiled-coil domain and SH3 domain serve as additional domains, but are also critical for FUT8 molecule assembly and maintenance of its enzyme activity. Based on molecular modeling and mutagenesis studies, it is believed that human FUT8 exerts its functions as the dimeric membrane-anchored glycosyltransferase in vivo [26]. Hydrophobic interactions involving a coiled-coil domain are required for the formation and maintenance of a homologous dimer. And the SH3 domain is near the coiled-coil domain in an inter-molecular manner. Also, according to Tomida et al.’s finding, FUT8 is partially localized on the cell surface, which is dependent on its SH3 domain. Additionally, ribophorin I has been identified as an SH3-dependent-binding protein of FUT8, which plays a crucial role in enhancing FUT8 activity [27].

### 2.3. Substrate Selection Preference for FUT8

Given the vast diversity and complexity of N-glycans and glycan branch structures, the substrate-specific recognition mechanism of FUT8 has captured many researchers’ attention. By testing the substrate preference of FUT8 to 77 different structures of N-glycans in absence of a peptide/protein, it was found that FUT8 has more strict identification requirements towards the α-(1,3)-mannose branch than the α-(1,6)-mannose branch. The catalytic reaction of core fucosylation can be performed only when the glycan α-1,3-mannose branch structure is modified with an agalactosylated and unprotected GlcNAC residue [28]. For bisecting N-glycans, FUT8 cannot be recognized and catalyzed. By resolving the crystal structure, it was found that the 4-position hydroxyl group of the first mannose of the N-glycan core structure was less than 5 Å away from the amino acid residue on the FUT8 protein, and the GlcNAc modification of the bisecting N-glycan on the 4-position hydroxyl group of mannose could produce an evident steric hindrance, and break the interaction [29]. Through analysis, it was found that GlcNAc on the arm of α-1,3 can form hydrogen bonds with FUT8 amino acid residues Asp^494^, Asp^495^, and His^535^. But it does not explain why FUT8 can catalyze core fucosylation of high-mannose N-glycans because there is no terminal GlcNAc group on its α-1,3 arm. According to Ana’s team, the protein environment surrounding N-glycans also influences FUT8 catalysis, and they hypothesized that certain key peptides could impact FUT8 recognition and binding [30]. Tseng et al. also studied the activity of FUT8 with three-antenna complex N-glycans, and found that the three-antenna glycan with GlcNAc terminated by N-acetylglucosamine transferase (GnT)-IV catalyzed [A3(2,4,2)] is the best substrate for FUT8, while the triantenna sugar [A3(2,2,6)] produced by GnT-V cannot be used as a substrate for FUT8, and when N-glycans are modified with core fucosylation, GnT-IV enzyme activity is inhibited, which reflects that there should be some rules of action between FUT8 and GnTs, but further exploration is required [31].

### 2.4. Role of Core Fucosylation Mediated by FUT8

Core fucosylation is estimated to occur in 20–90% of cell surface proteins, and contributes to the formation of sugar chains in cytokines, receptors, stem cells, lymphocytes, and other molecules, ultimately influencing cellular function [32,33,34,35]. FUT8, the sole glycosyltransferase responsible for N-glycan core fucosylation, is a type II transmembrane protein localized in the Golgi apparatus. GDP-fucose, the crucial building block for core fucosylation, is generated via two pathways: the de novo synthesis pathway and salvage pathway (Figure 2). Knockout of the FX protein, which blocks the de novo synthesis of GDP-fucose, resulted in death in most of the mice and postnatal dysplasia in survivors. However, oral L-fucose supplementation alleviated the disease, suggesting that the de novo synthesis pathway, which is the primary source, coexists with the salvage pathway [36]. Severe growth retardation and development delay, along with neurologic deficits and respiratory difficulties, have been reported in cases of loss-of-function mutations in FUT8 [37]. This is further supported by observations in mice with FUT8 knockout, which exhibit similar symptoms along with increased mortality rates [35]. Molecular mechanism studies suggested that these effects were associated with dysregulation of transforming growth factor-β1 (TGF-β1) receptor [35], epidermal growth factor receptor (EGFR) [38], α3β1integrin [39], and vascular endothelial growth factor receptor-2 (VEGFR-2) [34] activation and signaling due to loss of core fucosylation.

Additionally, dysregulation of core fucosylation mediated by FUT8 is implicated in several chronic diseases. In chronic obstructive pulmonary disease, impaired core fucosylation of the secreted protein acidic and rich in cysteine results in decreased collagen binding, disrupting cell–matrix communication and leading to abnormal alveolar structures [40], while elevated FUT8 expression would activate EGFR pathway and contribute to the development of psoriasis [41]. These discoveries highlight the critical role of FUT8 in preserving the normal physiological functions of the body.

**Figure 2 ijms-25-01068-f002:**
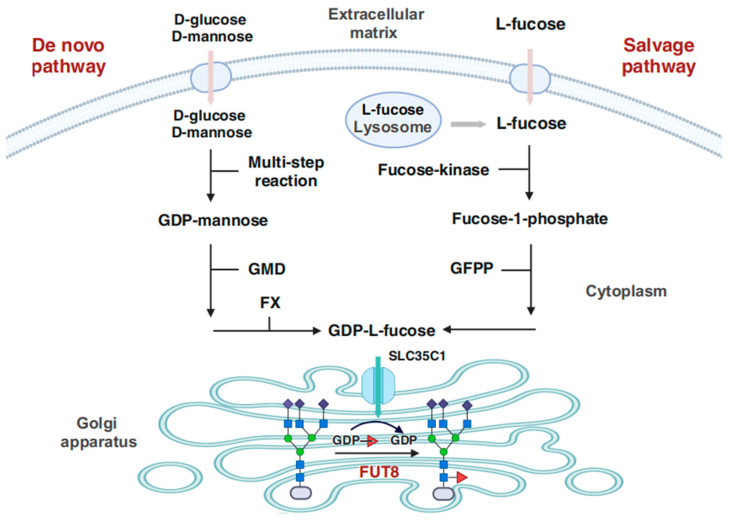
Synthesis pathway of FUT8 donor in humans. There are two synthetic pathways for GDP-fucose, which is produced in the cytoplasm and eventually utilized by FUT8 in the Golgi apparatus. The de novo pathway starts with D-mannose obtained by a multi-step enzymatic reaction of GDP-glucose, and then is converted to GDP-fucose by the interaction of GMDS and the FX protein [42,43]. The salvage pathway utilizes L-fucose from extracellular or lysosomal metabolism to synthesize GDP-fucose through a two-step reaction catalyzed by fucose-kinase and GFPP [44,45]. Abbreviations: GMDS, GDP-mannose-4,6-dehydrogenase, GFPP, GDP-fucose-pyrophosphorylase, FX, GDP-4-keto-6-deoxymannose-3,5-epimerase-4-reductase, Created with BioRender.com accessed on 18 February 2023.

## 3. FUT8 Is Widely Involved in Tumor Malignancy

Numerous studies have shown that FUT8 is abnormally overexpressed in various tumors such as melanoma, osteosarcoma, hepatocellular carcinoma (HCC), breast cancer, and prostate cancer. This overexpression significantly impacts the prognosis of patients and could be used as a biomarker for tumor diagnosis and a target for treatment intervention [46,47,48,49,50,51,52]. Remarkably, the dysregulated core fucosylation facilitated by FUT8 perturbs the expression of multiple molecules involved in tumor growth, metastasis, immunosuppression, etc. Consequently, it disrupts normal signaling pathways and plays a crucial role in malignant transformation [47,48].

With advancing research on relationships between FUT8 and tumors, there is a growing interest in the development of anti-cancer drugs targeting this enzyme. As a result, understanding the biological effects and regulation mechanisms of FUT8 is becoming a focal point of current research. Table 2 and Figure 3 show an overview of the functional effects and intrinsic molecular mechanisms about FUT8 in tumor development.

### 3.1. FUT8 and Metastasis

Cell adhesion molecules are closely connected with tumor-aggressive processes such as epithelial–mesenchymal transformation (EMT), tumor infiltration, and metastasis [75,76,77]. In addition to genetic and epigenetic alterations, mounting evidence suggests that post-translational dysregulation profoundly affects the function and activity of these adhesion molecules [78,79]. Daisuke Osumi et al. found that overexpression of FUT8 in human colon carcinoma WiDr cells contributes to the production of low-molecular-weight E-cadherin, leading to improved cell–cell adhesion and protein stability [55]. Lectin blot assays have also identified core fucosylation and N-glycosylation modifications on the low-molecular-weight E-cadherin. Notably, it displayed remarkable resistance to PNGase F.

EMT is a typical biological process that enables malignant tumor cells to acquire increased invasive and metastatic potential, and FUT8 has been reported to facilitate the EMT program through multiple channels. According to the findings of Cheng-Fen Tu’s team, FUT8 targets TGF-β RI and RII complexes, enhancing their ligand binding capacities. This leads to the activation of downstream signaling pathways and contributes to the aggressive behavior of breast cancer cells [50]. Furthermore, they recently demonstrated that core fucosylation is critical for integrin αvβ5- mediated cell adhesion on vitronectin and IL6ST responses, which involves EMT and metastasis [53]. A study employing mass spectrometry-based glycoproteomics observed precise changes in site-specific glycosylation during HGF/TGF-β1-induced EMT in HCC cells. It demonstrated that FUT8 plays a central role in driving HGF/TGF-β1-induced EMT. In detail, the core fucosylation of folate-receptor 1, especially at the glycosite Asn^201^, could strongly enhance folate uptake capacity [58].

In the above studies, EMT is accompanied by an increase in FUT8 expression, which in turn facilitates this cellular transformation. However, based on the study of Rubén López-Cortés’ team, a dynamic link between FUT8 and EMT in colorectal cancer (CRC) progression was indicated by diverse biological responses of SW480 and SW620 after inhibiting FUT8. Also, downregulation of core fucosylation may promote improved endothelial–epithelial interactions with potential carcinogenic effects [56]. Similarly, in breast cancer MCF-7 and MCF10A cells, FUT8 expression has no remarkable influence on E-cadherin [54,80]. These findings suggests that FUT8 may impact metastasis without directly interfering with EMT. And other regulatory mechanisms have been investigated. By detecting the changes of biomarkers related to invasion and migration, such as FAK, integrin-α3β1, β-catenin, and metalloproteinases (MMPs), it was finally proposed that FUT8 deficiency inhibits the migration of human breast cancer cells through three ways: inhibition of the FAK/integrin signaling pathway by suppressing the core fucosylation of E-cadherin; inhibition of Wnt/β-Catenin signaling by reducing the accumulation of nuclear β-catenin; inhibition of the deterioration of tumor microenvironment by down-regulating the expression of MMP-2 and MMP-9 [54]. In melanoma cells, FUT8 promotes invasion and tumor dissemination, partially due to impaired L1CAM cleavage caused by its aberrant core fucosylation [46]. During the tumor progression, aberrant expression of FUT8 often coincides with an increase in tumor metastasis, despite the diverse range of specific molecular mechanisms involved.

### 3.2. FUT8 and Immune Evasion

Tumor cells possess multiple strategies to evade immune recognition and attacks, and emerging studies have highlighted the crucial role of core fucosylation abnormalities in facilitating tumor immune evasion. As critical regulatory molecules in the immune system, immune checkpoint molecules are indispensable in maintaining self-tolerance and preventing autoimmunity. Unfortunately, cancer cells exploit these molecules to evade immune surveillance. Inhibitory checkpoint molecules, like programmed cell death protein 1 (PD-1), are overexpressed in exhausted T cells, hindering their activation [81]. Through CRISPR/Case9 genome-wide loss-of-function high-throughput screening, Masahiro Okada et al. identified FUT8 as a positive regulator of PD-1 expression. Knocking out FUT8 or treating cells with a GDP-fucose analogue could promote T cell proliferation, increase cytokine production (particularly IL-2 and IFN-γ), eliminate PD-1-related failures, and enhance anti-tumor efficacy [62]. Nianzhu Zhang et al. also confirmed in non-small cell lung cancer (NSCLC) that knockout of FUT8 can promote the ubiquitination and proteasome degradation pathway of PD-1, thereby enhancing the anti-tumor activity of cytotoxic T cells [63]. For ligand molecules, Yun Huang and colleagues have discovered a new immune checkpoint molecule, B7H3, which plays a role in triple-negative breast cancer where PD-1/PD-L1 immunotherapy has limited efficacy. Aberrant glycosylation mediated by FUT8 inhibits the proteasome-mediated ubiquitination degradation pathway to stabilize B7H3 expression on cell membranes. This inhibits the immune-killing effects of NK cells and T cells, leading to a poor prognosis [47], while, the instability of defucosylation B7H3 after FUT8 silencing in CRC is due to the binding of HSC70 to B7H3 triggering its degradation through the chaperone-mediated autophagy pathway [82]. In the context of head and neck squamous cell carcinoma (HNSCC), core fucosylation is a critical post-translational modification that stabilizes the expression of PD-L2 by blocking the ubiquitin-dependent lysosomal degradation pathway. This consequently enhances its binding to PD-1 and leads to immunosuppression. Furthermore, glycosylated PD-L2 would form a complex with EGFR, causing resistance to cetuximab [64].

Numerous inhibitors of immune checkpoint molecules have been developed to disrupt the binding between inhibitory receptors and ligands to enhance the anti-tumor effect. Prominent examples include approved humanized monoclonal antibodies such as Ipilimumab, Pembrolizumab, and Atezolizumab, which target CTLA4, PD-1, and PD-L1, respectively [83]. It is known that the presence of core fucosylation of immunoglobulin G antibodies strongly inhibits the antibody-dependent cell-mediated cytotoxicity [84]. In a study by Junko Takei et al., they developed an anti-PD-L1 monoclonal antibody called 13-mG2a-f with eliminated core fucosylation, which exhibits remarkable anti-tumor effects [85]. FUT8 mediates immunosuppression by enhancing the protein stability of immune checkpoint molecules. The combined blocking of the core fucoylation of immune checkpoint molecules and inhibitors is a potential way to improve therapeutic efficiency.

### 3.3. FUT8 and Other Malignant Phenotypes

The interdependence of tumor behaviors and the diverse effects of changes in critical components pose a challenge in accurately discerning their individual impacts. Beyond its eminent role in metastasis and immune evasion, FUT8 also influences other tumor behaviors, including proliferation, stemness, apoptosis, and tumorigenesis, although the degree of influence may vary across different types of tumors.

For example, FUT8 may cause radio resistance and poor prognosis by the targeted activation of CD147 in esophageal squamous cell carcinoma [68]. In castration-resistant prostate cancer, FUT8, serving as a crucial modulator, promotes the activation of EGFR and its downstream pathways, leading to increased resistance to androgen deprivation therapy [48]. In Huh7.5.1 hepatoma cells, FUT8 overexpression activates the PI3K/Akt/NF-κB signaling pathway, resulting in resistance to 5-FU [59]. Conversely, impairment of FUT8 in Ishikawa cells substantially inhibits cell proliferation [86]. Glioblastoma, characterized by high FUT8 expression, exhibits the regulation of malignant phenotypes such as MET through the activation of multiple receptor tyrosine kinases and their downstream signaling pathways [70]. In pancreatic cancer, often termed as the “king of cancer”, FUT8 correlates with cell proliferation, stemness, drug resistance, and tumorigenesis. Although the precise regulatory mechanism remains unclear, it has been demonstrated that FUT8 influences the expression and stability of EGFR but does not affect integrin β1 [73].

While FUT8 is generally overexpressed in most cancers, it exhibits a unique downregulation in osteosarcoma relative to normal tissues. This downregulation reduces the level of core fucosylation on tumor necrosis factor receptor (TNFR), leading to a decrease in Fas-associated death domain proteins and activation of the non-classical TNF/NF-κB signaling pathway, all of which contribute to lower patient survival rates [52]. In gastric cancer, FUT8 expression is also inhibited. The decrease in core fucosylation has been associated with enhanced malignancy, although the specific mechanism underlying this effect remains unclear [74].

### 3.4. FUT8 and Microenvironment Homeostasis

The close association between FUT8 and tumors is further evident in the changes of microenvironment components. Tumor-associated fibroblasts (CAFs), serving as a center for cross-communication between tumor stromal cells, play a crucial role in tumor progression. They secrete growth factors, inflammatory ligands, and extracellular matrix proteins that promote cancer cell proliferation, drug resistance, and immune evasion [87]. Research has revealed that FUT8 plays a critical role in the development of NSCLC by activating EGFR signaling to affect the cancer-promoting capability of CAFs [60]. Patients with cervical cancer often exhibit disrupted vaginal flora, leading to reduced levels of lactic acid bacteria that utilize carbohydrates in mucosal epithelial cells to produce lactic acid. This acid, in turn, limits the adhesion, colonization, and growth of pathogenic bacteria [88]. According to the studies of Qingjie Fan’s team, the levels of core fucose were significantly reduced in the serum of patients with cervical cancer, exfoliated cervical cancer cells, and tumor tissues. Knocking out FUT8 was found to promote the proliferation and invasion of cervical cancer cells, while the abundance of lactic acid bacteria positively correlated with the core fucosylation modification level [89]. Furthermore, research has revealed that lactic acid bacteria could activate the Wnt pathway through lactate–GPR81 complex metabolites. This eventually increases core fucosylation in vaginal mucosal epithelial cells while restraining the proliferation and migration of cervical cancer cells. Therefore, FUT8 is essential in maintaining the body’s microenvironment homeostasis.

## 4. Multilevel Regulation of FUT8 in Tumors

FUT8 is involved in regulating multiple tumor behaviors and is closely associated with malignant transformation. However, its impact varies across different cell types, underscoring the complexity of its research. Current reports mainly focus on the functional effects of FUT8, but investigating the underlying causes of aberrant expression and the detailed regulatory mechanisms is of great significance for early tumor diagnosis and the development of new therapeutic targets. In the following discussion, we delve into the existing research on the regulation of FUT8 expression (Table 3). And with HCC as the representative, Figure 4 illustrates the regulatory pathways of FUT8 based on published studies.

**Table 3 ijms-25-01068-t003:** Regulation factors of FUT8 expression in multiple types of tumors.

Tumor Type	Type of Factors	Factors	Effect on FUT8 Expression	Regulation Approach	Functional Effects on Tumor	Ref.
HCC	LncRNA	HOTAIR	Promotes	Recruits P300 to combine with STAT3, and the transcriptional complex co-modulated FUT8 and MUC1 expression	Activates JAK1/STAT3 cascade, promotes HCC progression	[57]
HCC	TF	β-catenin/TCF	Promotes	Binds to the FUT8 promoter region	Activates Wnt/catenin signaling, promote HCC progression	[90]
HCC	TF	Wild-type p53	Promotes	The activation of p53 via its acetylation evokes FUT8 expression	Enhances L-Fucose-Mediated Drug Delivery	[51]
HCC	Circular RNA	cFUT8	Promotes	Bind-free miR-548c in the cytoplasm, impair its inhibitory effect on FUT8 expression	Promotes proliferation and invasion, maintain malignant potential	[91]
HCC	Protein	Caveolin-1	Promotes	Activates Wnt/β-catenin signaling, which leads to downstream binding of the TCF/LEF to the FUT8 promoter region to activation of its transcription.	Enhances proliferation and invasion	[92]
Huh7.5.1 Cells	Virus	HCV	Promotes	Unknown	Promotes proliferation, causes 5-FU drug resistance	[59]
HepAD38 cell lines	Virus	HBV	Inhibits	Unknown	Unknown	[93]
HCC	MicroRNA	miR-122	Inhibits	Binds to the 3’UTR of FUT8	Unknown	[94]
HCC	MicroRNA	miR-34a	Inhibits	Binds to the 3’UTR of FUT8	Unknown	[94]
Melanoma	TF	TGIF2	Promotes	Binds to the FUT8 promoter region	Induces metastasis, contributes to melanoma aggressive behavior	[46]
Breast cancer	MicroRNA	miR-10b	Promotes	Binds to the 3’UTR of AP-2γ, which inhibit STAT3 phosphorylation	Activates the AKT pathway, enhances the motility and proliferation	[80,95]
Breast cancer	Opioid analgesics	Fentanyl	Promotes	Activates the Wnt/β-catenin signaling pathway	Induces stemness and EMT	[49]
NSCLC	MicroRNA	MiRNA-198-5p	Inhibits	Directly targets the 3′-UTR of FUT8 mRNA	Inhibits the migration, invasion, and EMT	[96]
LUAD	Circular RNA	cFUT8	Promotes	Competitively interacts with YTHDF2 and sponge miR-186-5p	Promotes the malignant progression	[97]
NSCLC	TF	β-Catenin/LEF-1	Promotes	Transcriptional activation	Promotes EMT	[98]
CRC	TF	Wild-type p53	Promotes	Binds to the FUT8 promoter region	FUT8 is associated with better DFS in tumors with negative p53	[99]
CRC	MicroRNA	miR-198	Inhibits	Targets the 3′UTR of FUT8	Suppresses proliferation and invasion	[100]
CRC	LncRNA	LEF1-AS1	Promotes	Recruits MLL1 to the promoter region of LEF1, induces H3K4me3 methylation modification and mediate LEF1 transcription	Activates Wnt/β-catenin pathway, promotes proliferation, migration, and invasion	[101]
OSCC	LncRNA	LncRNA	Promotes	Sponge miR-186, which is a direct target of FUT8	Regulate cell survival, apoptosis, and migration, a driver of OSCC progression	[102]
ICCA	MicroRNA	miR-122-5p	Inhibits	Binds to the 3’UTR of FUT8	Suppresses proliferation and migration	[69]
Pancreatic cancer	Nuclear receptor co-activator	NCOA3	Promotes	Unknown	Increases the stability of the mucins, contribute to the progression of pancreatic cancer	[71]

**Figure 4 ijms-25-01068-f004:**
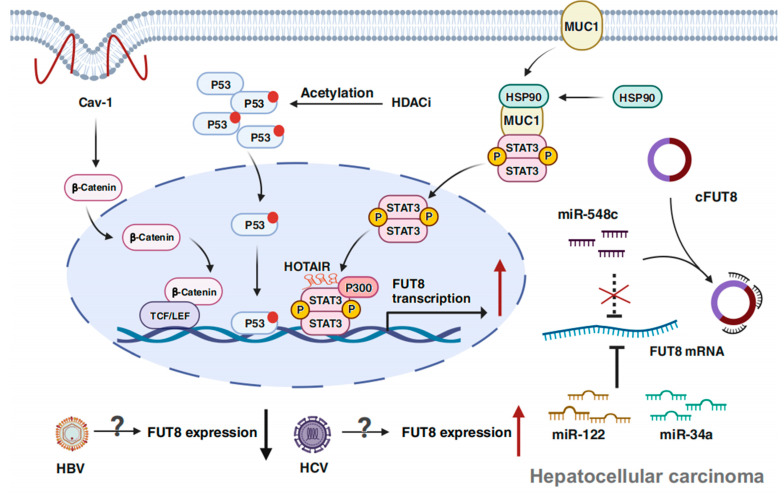
Diagram of the primary regulatory pathways of FUT8 in hepatocellular carcinoma. Overexpression of Cav-1 will activate β-catenin and promote its nuclear translocation, where it interacts with the TCF/LEF complex present at the FUT8 promoter region and ultimately enhances FUT8 transcription [92]. HDACi will induce acetylation of wild-type P53, resulting in transcriptional activation of FUT8 [51]. HSP90 can stabilize MUC1, and the combination of MUC1 and p-STAT3 will further activate JAK1/STAT3 cascade (not shown). Meanwhile, HOTAIR can recruit P300 to interact with STAT3, and boost FUT8 expression [57]. Free cFUT8 in the cytoplasm can adsorb miR-548c, which targets FUT8 3′UTR for degradation, indirectly upregulates FUT8 expression [91]. Some microRNAs, such as miR-122 and miR-34a, can inhibit FUT8 expression by targeting its 3 ‘UTR [94]. HCV and HBV have also been reported to affect FUT8 expression, but the precise molecular mechanism remains to be clarified [59,93]. Abbreviations: Cav-1, caveolin-1; cFUT8, circular RNA cFUT8; HDACi, histone deacetylase inhibitor. Created with BioRender.com accessed on 18 February 2023.

### 4.1. At the Transcriptional Level

Transcription factors act as the initial checkpoint in DNA decoding sequences, operating as master regulators and gene selectors. They are responsible for cell-type determination, development patterns, specific pathway control, and more. Therefore, identifying the transcription factors that act on the FUT8 gene can help us understand its expression patterns and shed light on the underlying reasons behind its broader biological functions.

P53, a transcription factor-encoding gene, is involved in regulating multiple signaling pathways and various cellular processes, including cell cycle arrest, DNA repair, apoptosis, aging, and metabolism [61]. Mutations in P53 have been observed in more than 50% of cancer, giving rise to a loss of its tumor suppressor function and promoting carcinogenesis [103]. Yutaka Okagawa et al. reported that wild-type P53 can bind to the FUT8 promoter region, leading to transcriptional activation [51]. Subsequently, Masaru Noda’s team studied the effect of P53 status on FUT8 prognosis in colorectal cancer [99]. Their study revealed that tumors with wild-type P53 exhibited higher levels of FUT8 mRNA compared to those with P53 mutations. However, no statistically significant difference was found regarding the impact of FUT8 expression and P53 status on disease-free survival (DFS) among stage II and III colorectal cancer patients. Only in the wild-type P53 tumor group did FUT8-positive expression correlate with improved DFS. This report indicates that the prognostic value of FUT8 expression depends on P53 status. Nevertheless, further extensive studies are required to draw a concrete conclusion.

Yanru Guo’s team conducted a study on liver cancer, revealing that a high level of FUT8 expression leads to an enhanced core fucosylation of Hsp90. This augmentation promotes the stabilization of the binding between MUC1 and p-STAT3, triggering the activation of the JAK1/STAT3 signaling cascade, thereby participating in the progression of liver cancer. They subsequently unveiled the involvement of epigenetics, where HOTAIR was found to recruit P300. This recruitment leads to the formation of a transcriptional complex with p-STAT3, collectively responsible for regulating the expression of FUT8 [57]. In another study conducted by Minxing Ma et al. in breast cancer cells, the transcription factor activating protein 2γ (AP-2γ) was identified as capable of binding to STAT3, thereby reducing the production of p-STAT3. However, miR-10b, which is highly expressed, counteracts the inhibitory effect of AP-2γ on p-STAT3, resulting in the increased FUT8 expression and proliferation of malignant cells [95].

In NSCLC, FUT8 is highly expressed in the process of EMT. Researchers have utilized computational prediction and in vitro verification to identify that the binding site of FUT8 5’UTR contains LEF-1/TCF. It is proposed that FUT8 may contribute to the progression of lung cancer via a positive feedback pathway: the EMT-induced downregulation of E-cadherin leads to nuclear accumulation of β-catenin. Consequently, β-catenin facilitates the LEF-1-mediated activation of FUT8 expression, thereby promoting NSCLC progression [98]. Cheng et al. discovered in mouse hepatoma cells that Caveolin-1 (Cav-1) activates the Wnt/β-catenin signaling pathway, resulting in the increased expression of FUT8. This, in turn, promotes cell proliferation and invasion. Similarly, this process is facilitated by LEF-1/TCF binding to the promoter region of FUT8 [92].

### 4.2. At the Post-Transcriptional Level

Advancement in high-throughput RNA sequencing and systems biology has greatly deepened our understanding of the critical role of non-coding RNA (ncRNA) in tumor. They have been associated with a wide range of processes, including but not limited to carcinogenesis, inhibition, apoptosis, stemness, and drug resistance [104]. Studies have suggested that ncRNA can modulate the expression of FUT8, leading to either inhibition or promotion.

Chong Li et al. observed that circular RNA cFUT8 and FUT8 alike were upregulated in HCC tissues, with similar impact on tumor characteristics. Specifically, both cFUT8 and FUT8 promote HCC proliferation, invasion, tumorigenic potential, and are linked to EMT. Mechanically, the presence of miR-548c inhibits FUT8 expression, but cFUT8 alleviates this inhibition by regulating miR-548c’s binding to FUT8 in the cytoplasm [91]. Also in HCC, Lei Cheng’s team identified miR-26a, miR-34a, and miR-455-3p that can target FUT8 through microRNA microarrays [105]. These microRNAs bind to the 3’ UTR region of FUT8 and downregulate its expression. In NSCLC, miR-198-5p possesses a cancer inhibitory function by directly targeting and inhibiting FUT8 expression, thereby impeding EMT. However, FUT8 overexpression can reverse the suppressive effect of miR-198-5p [96]. In a recent report on lung adenocarcinoma, m6A-modified cFUT8 stabilizes FUT8 expression by competitively interacting with YTHDF2. Also, it is able to function as a sponge for miR-186-5p to increase FUT8 expression [97]. In oral squamous cell carcinoma (OSCC), miR-186 has been discovered to directly target and inhibit the expression of FUT8. Nevertheless, a highly expressed lncRNA SNHG1 functions as a sponge for miR-186, counteracting its inhibitory effect and promoting the malignant growth of OSCC [102]. And miR-198 has been found to act directly on the 3′UTR of FUT8 in CRC, thereby reducing FUT8 expression and inhibiting cell proliferation, migration, and invasion [100]. In intrahepatic cholangiocarcinoma, highly expressed FUT8 promotes proliferation and migration by activating the PI3K/AKT signaling. MiR-122-5p can inhibit the expression of FUT8, but its expression is low in cancer [69]. While Mengjiao Zhang et al. explained the mechanism of the reduction of tumor-suppressive miR-122-5p in CRC infected by fusobacterium nucleatum, this infection would trigger the secretion of intracellular miR-122-5p into exosomes via an RNA-binding protein-hnRNPA2B1 [106].

LEF1-AS1, a lncRNA transcribed from the opposite direction of the LEF1 promoter region, is highly expressed in multiple tumors. Yu Qi et al. found that in CRC, LEF1-AS1 may regulate LEF1 expression by recruiting the MLL1-mediated methylation modification of H3K4me3 at the LEF1 transcription start site. Moreover, LEF1 influences FUT8 expression while LEF1-AS1 acts as a positive regulator of CRC progression. Finally, the authors proposed that the LEF1-AS1/LEF1/FUT8 axis facilitates CRC progression by activation of the Wnt/β-Catenin pathway [101]. In melanoma, FUT8-AS1, also known as FUT8 antisense RNA1, promotes the biosynthesis of miR-145-5p and inhibits the NRAS/MAPK signaling pathway, exerting an anti-cancer effect [107]. Conversely, FUT8 exhibits a pro-carcinogenic activation effect and is markedly upregulated in melanoma metastases [46]. The contradictory effects of FUT8-AS1 and FUT8 on the same cancer type are intriguing. Despite the potential significance of FUT8-AS1, research remains limited, with most investigations focusing on bioinformatic predictions, and its molecular mechanisms in pan-carcinoma remain unclear.

### 4.3. Impact of Environmental Factors

Recent studies suggest that environmental factors, in addition to conventional transcription factors and epigenetic modifications, can significantly modulate FUT8 expression in vitro. For instance, hepatitis C virus infection with human liver cancer line Huh7.5.1 leads to upregulated FUT8 expression, resulting in cell proliferation and drug resistance [59]. In contrast, expression of FUT8 is down-regulated after infection with the hepatitis B virus in HepAD38 cells [93]. Metabolites from lactic acid bacteria in the vaginal environment can activate FUT8′s transcription and enhance core fucosylation in mucosal epithelial cells, thereby inhibiting the proliferation and invasion of cervical cancer cells [89]. In clinical cancer treatment, chemotherapy and adjuvant analgesia are commonly used strategies, but some adjuvant drugs may inadvertently promote the activation of tumor cells, leading to unfavorable outcomes for patients. Yang et al. found that fentanyl activates the Wnt/β-catenin signaling pathway to regulate FUT8 expression, in turn, inducing stemness and EMT in breast cancer cells [49]. The discovery of these influential environmental factors provides promising insights for the development of effective cancer treatment strategies.

## 5. Concluding Remarks and Future Perspectives

Collectively, mounting evidence strongly supports that aberrant FUT8 expression is a critical driver of tumorigenesis and progression. In contrast to normal tissues, tumors exhibit significant alterations in FUT8 expression, which can have a profound impact on patient survival and serve as a valuable diagnostic tool in clinical settings. Of course, the diagnostic potential of FUT8 extends beyond its own abnormal expression. Furthermore, abnormal core fucosylation, which is regulated by FUT8 in glycoproteins, shows promise as a valuable clinical diagnostic marker. For example, the degree of core fucosylation in AFP has been shown to be a more accurate indicator of HCC progression [108], while serum fucosylated haptoglobin holds potential as a promising prognostic biomarker for prostate cancer, outperforming prostate-specific antigen in predicting Gleason score upgrades [109].

Given the prominent effect of FUT8 on tumors, it has been considered as an attractive target for tumor therapy. However, research on selective inhibitors of FUT8 has been hampered for decades by limited information on the specific structure about FUT8 and its substrate complex [110]. Some L-fucose analogues that competitively inhibit FUT8 activity through interference with the de novo synthesis of GDP-fucose have been reported in cancer studies. For instance, the treatment of 2-fluoro-L-fucose effectively and significantly inhibits the proliferation, migration, and tumorigenicity of human hepatocellular carcinoma HepG2 cells [111]. In a first-in-human, first-in-class, phase I study in advanced solid tumors, SNG-2FF showed excellent anti-tumor effects in many types of cancer. While it is associated with thromboembolic events [112]. The disadvantages of these analogs, such as poor stability and high polarity, further limit their application. Moreover, all human fucosyltransferases share the same substrate; these analogues cannot selectively inhibit FUT8 and may invite severe toxic side effects. Happily, a novel inhibitor targeting FUT8 independent of GDP-fucose, FDW028, has recently been proposed for the first time, showing outstanding specificity and significant anti-tumor effects in CRC [82]. Meanwhile, effective strategies to target and inhibit the core fucosylation specifically in critical glycoproteins, such as EGFR and L1CAM, based on tumor characteristics, are necessary and essential. This may be a direction for the precision treatment of tumors.

Currently, more research is concentrated on analyzing the effects of FUT8 on various tumor functions and its internal regulatory mechanism of self-expression, which could help identify more precise targets and develop innovative strategies to address cancer-related issues. The continual progress of science and technology offers promising opportunities, with advanced methods such as matrix-assisted laser desorption/ionization—mass spectrometry imaging (MALDI-MSI) for detecting sugar chain alterations [113] and high-throughput library screening for identifying target genes [114]. With continued studies into the role of FUT8 in cancer, we are confident that more effective and comprehensive cancer treatment strategies that benefit humanity will soon be developed.

## Figures and Tables

**Figure 1 ijms-25-01068-f001:**
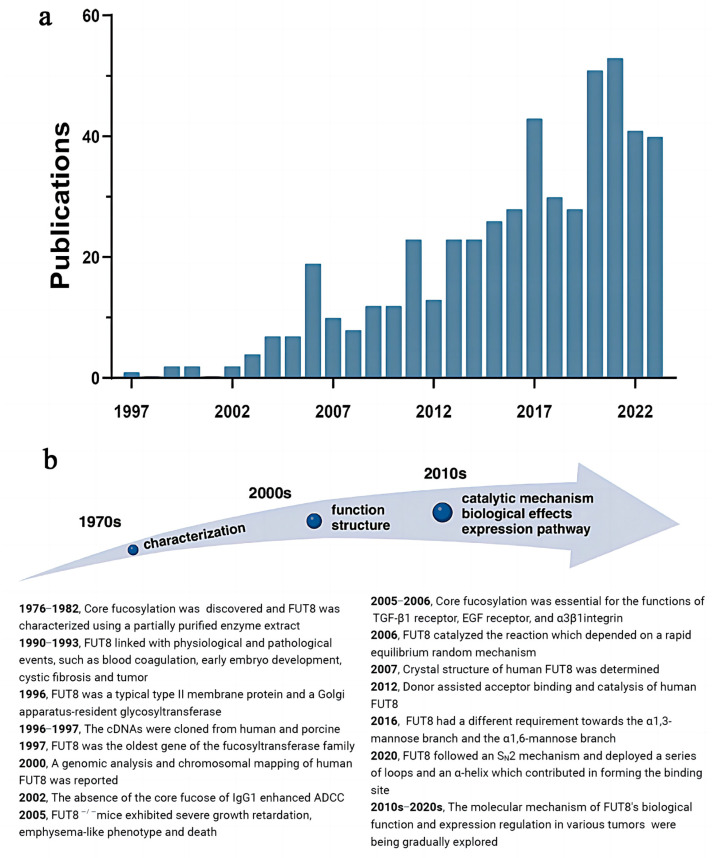
The primary research process for FUT8. (**a**) Statistics on the number of FUT8-related articles in the last 20 years from the Web of Science. The number of published articles has been steadily increasing, which reflects people’s growing interest in FUT8 research, indicating the significant potential value in exploring FUT8′s applications. (**b**) A timeline of the major advances in FUT8 research. The key discoveries regarding FUT8 have been summarized since its discovery. It is evident that FUT8 plays a crucial role in promoting tumor progression, and its regulatory mechanism is becoming a focus of intensive research. Created with BioRender.com accessed on 18 February 2023.

**Figure 3 ijms-25-01068-f003:**
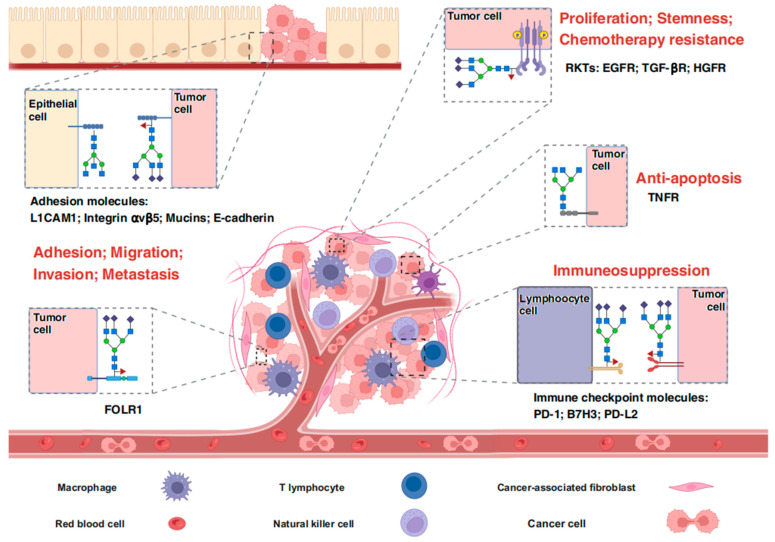
Proteins interacting with FUT8 and their biological effects in tumor progression. Aberrant core fucosylation mediated by FUT8 alters the biological function of key proteins, induces excessive activation of tumor signaling pathways, and promotes malignant transformation of tumors, such as proliferation, metastasis, and immunosuppression. Created with BioRender.com accessed on 18 February 2023.

**Table 1 ijms-25-01068-t001:** An overview of known fructosyltransferases in humans.

Name	Abbreviation	Type of Fucosylation	Function	Subcellular Location	Ref.
Galactoside α-(1,2)-fucosyltransferase 1	FUT1	α-(1,2)-fucosylation	H antigen	Golgi apparatus	[4]
Galactoside α-(1,2)-fucosyltransferase 2	FUT2	α-(1,2)-fucosylation	H antigen	Golgi apparatus	[5]
3-galactosyl-N-acetylglucosaminide 4-α-L-fucosyltransferase	FUT3	α-(1,3)/α-(1,4)- fucosylation	Lewis x/y/a/b, Sialyl Lewis x/a	Golgi apparatus, extracellular	[6,7,8]
α-(1,3)-fucosyltransferase 4	FUT4	α-(1,3)-fucosylation	Lewis x antigen	Golgi apparatus, Golgi membrane	[6,9]
4-galactosyl-N-acetylglucosaminide 3-α-L-fucosyltransferase FUT5	FUT5	α-(1,3)-fucosylation	Lewis x, Sialyl Lewis x antigen	Golgi apparatus, extracellular	[10,11]
4-galactosyl-N-acetylglucosaminide 3-α-L-fucosyltransferase FUT6	FUT6	α-(1,3)-fucosylation	Lewis x, Sialyl Lewis x antigen	Golgi apparatus, extracellular	[10,12]
α-(1,3)-fucosyltransferase 7	FUT7	α-(1,3)-fucosylation	Sialyl Lewis x antigen	Golgi apparatus	[6]
α-(1,6)-fucosyltransferase	FUT8	α-(1,6)-fucosylation	Core fucosylation	Golgi apparatus	[3]
4-galactosyl-N-acetylglucosaminide 3-α-L-fucosyltransferase 9	FUT9	α-(1,3)/α-1-4-fucosylation	Lewis x/y/a/antigen	Golgi apparatus, endoplasmic reticulum	[13]
α-(1,3)-fucosyltransferase 10	FUT10	α-(1,3)/α-1-4-Fucosylation	Unknown	Golgi apparatus, endoplasmic reticulum, nucleoplasm	[14]
α-(1,3)-fucosyltransferase 11	FUT11	α-(1,3)/α-1-4-fucosylation	Unknown	Golgi apparatus, endoplasmic reticulum, nucleus	[14]
GDP-fucose protein O-fucosyltransferase 1	POFUT1	Peptide-O-Fucosyltransferase 1	Fucosyl protein	Endoplasmic reticulum	[15]
GDP-fucose protein O-fucosyltransferase 2	POFUT2	Peptide-O-Fucosyltransferase 2	Fucosyl protein	Endoplasmic reticulum	[16]

**Table 2 ijms-25-01068-t002:** Biological effects and mechanism of abnormal expression of FUT8 in different tumors.

Tumor Type	Compared to Normal Levels	Targets	Mechanism	Function	Ref.
Breast cancer	Elevated	TGF-β receptor I and II	Activate TGF-β1 signaling	Stimulate EMT process	[50]
Breast cancer	Elevated	Integrin αvβ5/IL6ST	Enhances responsiveness to IL-6 or OSM signaling	Promotes EMT and metastasis	[53]
Breast cancer	Elevated	E-cadherin	Inhibits the FAK/integrin pathway, increase nuclear β-catenin accumulation and expressions of MMP-2 and MMP-9 after FUT8 deficiency	FUT8 deficiency suppresses adhesion, invasion, and migration	[54]
TNBC	Elevated	B7H3	Increases protein stability and cell-surface expression	Promotes immune evasion	[47]
WiDr human colon carcinoma cells	Elevated	E-cadherin	Unknown	Enhances cell–cell adhesion	[55]
The epithelial-like non-metastatic SW480 line	Elevated	E-cadherin	Unknown	Shows greater growth capacity and adhesion to EA.hy926 cells, decreases migration after silencing FUT8	[56]
The metastatic fibroblast-like SW620 line	Elevated	E-cadherin	Unknown	Shows no significant changes in adhesion and migration, inhibits proliferation after FUT8 downregulation	[56]
HCC	Elevated	HSP90/MUC1	Enhances the binding of Hsp90 and MUC1, activate JAK1/STAT3 cascade	Promotes proliferation, aggression, and tumorigenesis	[57]
HCC	Elevated	FOLR1	Enhances the folate uptake capacity	Promotes the EMT	[58]
HCV-infected Huh7.5.1 cells	Elevated	Unknown	Activate PI3K-AKT-NF-κB signaling, induce the expression of P-gp, MRP1	Promote proliferation, cause chemotherapy-resistance	[59]
HCC	Elevated	Unknown	Activates PI3K/Akt signaling pathway	Contributes multidrug resistance	[11]
HCC	Elevated	GP73	Independent of cellular fucosylation, but dependent on overexpression of FUT8 in the Golgi apparatus	Increases the expression of GP73, a potential tumor marker for HCC	[12]
NSCLC	Elevated	SOD3	Sustains secretion and enzymatic activity of SOD3	Contributes to the suppression of cell growth of NSCLC cells	[13]
NSCLC	Elevated	Unknown	Inhibition of FUTs attenuates the TGF-β mediated phosphorylation of Smad2/3 and translocation of Smad3 into the nucleus	Inhibition of FUTs attenuates TGF-β-induced cell migration and tumor metastasis	[14]
NSCLC	Elevated	CAFs and EGFR	Activates the EGFR signaling	Maintains cancer-promoting capacity of CAFs, contribute to the construction of an invasive TME	[60]
NSCLC	Elevated	Unknown	Unknown	Enhances tumor metastasis and growth, is correlated with poor OS and DFS of patients	[61]
LUAD	Elevated	PD-1	Attenuate the PD-1 ubiquitination	Inhibit CTL activation and anti-tumor responses.	[62,63]
Lung cancer *	Elevated	E-cadherin	Activate Src/AKT signaling, mediate β-catenin phosphorylation and nuclear accumulation	Induce EMT-like process, stimulate migration	[15,16]
HNSCC	Elevated	PD-L2	Stabilizes PD- L2 by blocking ubiquitin-dependent lysosomal degradation, maintains the formation of PD- L2/EGFR complex, activates EGFR/STAT3 signaling, decreases the cetuximab binding affinity to EGFR	Promotes immune evasion and malignancy	[64]
Prostate cancer	Elevated	Extracellular vesicles	Unknown	Reduce the number of vesicles released into the extracellular space, increase the abundance of proteins associated with cell motility and metastasis	[65]
Androgen-resistant LAPC4 cells	Elevated	Unknown	Unknown	Reduce the production of PSA, drive castration resistance	[66]
CRPC	Elevated	EGFR	Involves the transformation of prostate cancer cells from nuclear receptor AR-dependent signaling to the cell surface including the EGFR signaling	Promotes DNA replication and cell cycle progression, rescue prostate cancer cells from depleted androgen-induced cell death	[48]
Melanoma	Elevated	L1CAM	Inhibits the cleavage of L1CAM, which is a mediator of the pro-invasive effects of FUT8	Facilitates invasion, tumor dissemination, and metastasis	[46]
EOC	Elevated	CTR1	Activation of JNK and ERK signaling	Decreases the ability of cisplatin uptake, leads to drug-resistance	[67]
ESCC	Elevated	CD147	Unknown	Confers radio resistance and a poor prognosis	[68]
ICCA	Elevated	Unknown	Activates the PI3K/AKT pathway	Promotes cell proliferation, migration, and invasion	[69]
Glioblastoma cells	Elevated	EGFR/HGF receptor	Activates multiple RTK signaling, especially the HGF/MET pathway	Contributes to the malignant behaviors and temozolomide therapy	[70]
Pancreatic cancer	Elevated	Mucins	Stabilize mucins post translationally	Promote the progression of tumor	[71]
PDAC	Elevated	Unknown	Unknown	Promote invasion and metastasis	[72]
Pancreatic carcinoma	Elevated	Unknown	Unknown	Promote proliferation, migration, stemness and chemoresistance	[73]
Gastric cancer	Attenuated	Unknown	Unknown	Contribute to Malignancy	[74]
Osteosarcoma	Attenuated	TNFRs	Activates the non-canonical NF-κB signaling pathway	Decreases mitochondria-dependent apoptosis, result in the poor OS of patients	[52]

Abbreviations: CTR1, copper transporter 1; EOC, epithelial ovarian cancer; ESCC esophageal squamous cell carcinoma; GP73, Golgi phosphoprotein 2; HGF, hepatocyte growth factor; HNSCC, head and neck squamous cell carcinoma; ICCA, intrahepatic cholangiocarcinoma; MRP1, multidrug resistance-associated protein; PDAC, pancreatic ductal adenocarcinoma; P-gp, permeability glycoprotein; PSA, prostate specific antigen; RTK, receptor tyrosine kinase; SOD3, superoxide dismutase 3; TNFR, tumor necrosis factor receptor; * Although FUT8 expression level was upregulated overall in lung cancer, the core fucoylation level of E-cadherin was decreased in tumor tissue.

## Data Availability

No new data were created or analyzed in this study. Data sharing is not applicable to this article.

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
