# Peer review of "The Multifaceted Role of FUT8 in Tumorigenesis: From Pathways to Potential Clinical Applications"

_ijms, 2024, doi:10.3390/ijms25021068_

Round 1
Reviewer 1 Report
Comments and Suggestions for Authors
This review highlights the significant role of FUT8, in tumorigenesis and development. The text underscores the crucial impact of aberrant FUT8 expression on cellular components, disrupting tumor signaling pathways and contributing to malignant transformations like proliferation, invasion, metastasis, and immunosuppression. The mention of FUT8's association with unfavorable outcomes in various tumors emphasizes its potential as a diagnostic marker.
In my opinion, there is a lack of information about clinical trials.
Author Response
Thank you very much for taking the time to review this manuscript. Please find the detailed responses below and the corresponding revisions highlighted changes in the re-submitted files.

Reviewer 2 Report
Comments and Suggestions for Authors
The manuscript, titled “The Multifaceted Role of FUT8 in Tumorigenesis: From Pathways to Potential Clinical Applications”, has comprehensively summarized the biological effects and regulatory mechanisms of FUT8 in tumors. Additionally, the biological characteristics of FUT8 and its clinical research value were introduced and discussed. Overall, I think that this article is logical, illustrated, and readable, but there are some minor problems that need to be corrected.
1. It is recommended to add references to the description of the FUT8 regulatory pathways in the figure legend of Figure 4.
2. It is suggested to increase much more the latest literatures in 2023.
Comments on the Quality of English LanguageMinor editing of English language required
Author Response

(The authors gave the same response as above.)
